# Genome-Wide Identification and Characterization of the PHT1 Gene Family and Its Response to Mycorrhizal Symbiosis in *Salvia miltiorrhiza* under Phosphate Stress

**DOI:** 10.3390/genes15050589

**Published:** 2024-05-06

**Authors:** Xue Chen, Yanhong Bai, Yanan Lin, Hongyan Liu, Fengxia Han, Hui Chang, Menglin Li, Qian Liu

**Affiliations:** 1College of Pharmacy, Shandong University of Traditional Chinese Medicine, Jinan 250355, China; xchenaylli@sina.com (X.C.); white_byh1122@163.com (Y.B.); 18754621424@163.com (Y.L.); 15554876110@163.com (F.H.); 17861171650@163.com (M.L.); 2Experimental Center, Shandong University of Traditional Chinese Medicine, Jinan 250355, China; lhyan0@163.com; 3Innovative Institute of Chinese Medicine and Pharmacy, Shandong University of Traditional Chinese Medicine, Jinan 250355, China; ch0194@126.com

**Keywords:** *Salvia miltiorrhiza*, phosphate transporter 1 (PHT1), arbuscular mycorrhizal fungi (AMF), phosphate stress

## Abstract

Phosphorus (P) is a vital nutrient element that is essential for plant growth and development, and arbuscular mycorrhizal fungi (AMF) can significantly enhance P absorption. The phosphate transporter protein 1 (PHT1) family mediates the uptake of P in plants. However, the *PHT1* gene has not yet been characterized in *Salvia miltiorrhiza*. In this study, to gain insight into the functional divergence of *PHT1* genes, nine *SmPHT1* genes were identified in the *S. miltiorrhiza* genome database via bioinformatics tools. Phylogenetic analysis revealed that the PHT1 proteins of *S. miltiorrhiza*, *Arabidopsis thaliana*, and *Oryza sativa* could be divided into three groups. PHT1 in the same clade has a similar gene structure and motif, suggesting that the features of each clade are relatively conserved. Further tissue expression analysis revealed that *SmPHT1* was expressed mainly in the roots and stems. In addition, phenotypic changes, P content, and *PHT1* gene expression were analyzed in *S. miltiorrhiza* plants inoculated with AMF under different P conditions (0 mM, 0.1 mM, and 10 mM). P stress and AMF significantly affected the growth and P accumulation of *S. miltiorrhiza*. *SmPHT1;6* was strongly expressed in the roots colonized by AMF, implying that *SmPHT1;6* was a specific AMF-inducible PHT1. Taken together, these results provide new insights into the functional divergence and genetic redundancy of the *PHT1* genes in response to P stress and AMF symbiosis in *S. miltiorrhiza*.

## 1. Introduction

Phosphorus (P) is one of the essential mineral nutrient elements required for plant growth and development and plays an important role in processes such as energy storage and transfer, signal transduction, enzyme regulation, and gene expression [1]. Phosphorus primarily exists in the soil in complex, insoluble organic forms that are inaccessible to plants. The content of inorganic phosphorus is relatively low; when the soil pH exceeds 7.0, inorganic phosphorus is predominantly mineralized and immobilized as calcium phosphates [2,3]. The diffusion rate of soil phosphates is much lower than the absorption rate of phosphates by plant roots, thereby limiting the P supply available to plants. Low P utilization efficiency has long been recognized as a key limiting factor for crop growth worldwide [4]. Thus, to enhance P utilization efficiency and increase crop yield, it is imperative to elucidate the strategies and associated mechanisms for improving P acquisition and utilization.

To address the issue of P deficiency in soil, plants have evolved a variety of strategies, such as increasing the root–soil contact area and establishing symbiotic relationships with arbuscular mycorrhizal fungi (AMF). AMF are widely distributed, as they are capable of forming symbiotic associations with more than 80% of terrestrial higher plants [5,6]. This symbiotic relationship aids in the absorption of mineral nutrients from the soil, particularly P [3]. AMF extract P from the soil containing the rhizosphere by extending their extraradical mycelia away from the root surface [5]. Subsequently, P is transported along fungal hyphae via the mycorrhizal pathway to the specialized symbiotic interface within the root cortex in exchange for plant carbon resources [7]. Furthermore, when plants form a symbiotic relationship with AMF, they can obtain up to 90% of their P requirements from the AMF [8].

The absorption and transport of P in plants are typically mediated by phosphate transporters (PHTs). The PHT gene family has been identified in plant species such as *Sorghum* [9], *Solanum tuberosum* [10], *Malus domestica* [11], and *Liriodendron tulipifera* [12] through whole-genome analysis. Based on the sequence characteristics and subcellular localization, members of the PHT family can be classified into five types: PHT1, located in the plasma membrane; PHT2, located in chloroplasts; PHT3, located in mitochondria; PHT4, located in chloroplasts, Golgi apparatus, and non-photosynthetic plastids; and PHT5, located in vacuoles [13,14,15]. Among these proteins, PHT1, which belongs to the major facilitator superfamily, plays an important role in the acquisition of inorganic phosphate from soil and its transport within plants [16]. The PHT1 family has been widely studied in plants. In the model plants *Arabidopsis thaliana* and *Oryza sativa*, a total of 9 and 13 *PHT1* genes, respectively, have been identified [17,18]. Furthermore, multiple P-transport proteins from the PHT1 family induced by AMF have been identified and investigated. For example, *NaPT5* expression in roots is triggered by AMF in *Nicotiana attenuate* [19]. In *Lotus japonicus*, *LjPT4* was expressed in arbusculated plant cells and in the root tips of non-mycorrhizal plants, and impairment of *LjPT4* reduced the response of roots to P [20]. The PHT1 gene family exhibited similar expression patterns in *Capsicum frutescens*, *Solanum melongena*, and *Nicotiana tabacum*. The expression of *PHT1;1* and *PHT1;2* was upregulated in response to P deficiency, but their expression levels decreased after inoculation with AMF. *PHT1;3* expression increases with mycorrhizal colonization, while *PHT1;4* and *PHT1;5* are specifically expressed in response to mycorrhizal symbiosis under low-P conditions [21]. Although some PHT1 members have been functionally studied in plants, a comprehensive analysis of PHT1 proteins in *Salvia miltiorrhiza* is lacking.

*S. miltiorrhiza* is well-known for its use in traditional Chinese medicine and is considered a model organism for medicinal plant research [22,23,24]. It is mainly used for the treatment of cardiovascular and cerebrovascular diseases. Phenolic acids, as effective components of *S. miltiorrhiza*, exhibit various biological properties, such as antioxidative, anti-inflammatory, antibacterial, antitumor, and cardioprotective effects [25]. Previous studies have demonstrated that AMF symbiosis can increase the accumulation of P and promote the synthesis of phenolic acids. However, the regulatory mechanism underlying this phenomenon still requires further investigation [26].

In this study, a total of nine *SmPHT1* genes were identified from the genome of *S. miltiorrhiza*. The characteristics of these genes were analyzed using bioinformatics and experimental methods. The gene expression patterns in response to different P levels (0 mM, 0.1 mM, and 10 mM) during AMF colonization were investigated via RT-qPCR. This study provides a foundation for further research on the functional roles of *SmPHT1* genes.

## 2. Materials and Methods

### 2.1. AMF Inoculum Preparation, Plant Materials, and Stress Treatments

The original inocula of AMF *Rhizophagus irregularis* was cultivated through symbiosis with scallion plants. The river sand used in the study after rinsing with clean water 20 times was mixed with perlite at a 1:1 ratio and autoclaved (121 °C, 1 h). Scallion seeds were sown in plastic pots filled with sterilized sand and perlite (1:1 ratio) and moderate AMF original inocula. After about two months of cultivation, the mycorrhizal colonization rate was detected, and inoculum was collected from pot cultures of scallion plants. The inoculum was a mixture of hyphae, spores, root fragments, sand, and perlite that contained approximately 800 spores per 100 g.

The *S. miltiorrhiza* cultivar ‘Huadan 2’ used in this study was planted in the Shandong University of Traditional Chinese Medicine Medicinal Botanical Garden [27]. One-month-old uniform *S. miltiorrhiza* seedlings were transplanted into pots (1 L volume) filled with sterilized sand and perlite at a 1:1 ratio. For the mycorrhizal treatments, 50 g of *Rhizophagus irregularis* inoculum was added to each pot. An equal amount of autoclaved (121 °C, 1 h) inoculum was added to the non-AMF culture pot.

In the experiments, three P concentrations (0 mM, 0.1 mM, and 10 mM) were used, and each treatment was replicated in ten pots. The experiments were repeated in triplicate. The *S. miltiorrhiza* plants were watered with modified Hoagland solution with different P levels once every two weeks. The plants were grown in a greenhouse under a 16 h light/8 h dark cycle at 23 °C and a relative humidity of 70%. After three months of cultivation, root samples were harvested and one part was used to determine the fresh weight, P concentration, and AMF colonization rate, and an additional part was preserved via rapid freezing in liquid nitrogen at −80 °C for RNA extraction. Total P content was determined using the molybdenum antimony anticolorimetric method [28]. To analyze tissue-specific gene expression, roots, stems, leaves, and flowers were collected from two-year-old *S. miltiorrhiza* plants at the Shandong University of Traditional Chinese Medicine Medicinal Botanical Garden [27].

### 2.2. Identification and Phylogenetic Analysis of SmPHT Genes in S. miltiorrhiza

The amino acid sequences of PHT1 proteins in *A. thaliana* were used as query probes to search the *S. miltiorrhiza* genome via BLASTP. Furthermore, the conserved domains of the *SmPHT1* proteins were analyzed using the Conserved Domain Database (CDD; https://www.ncbi.nlm.nih.gov/Structure/cdd/wrpsb.cgi, accessed on 1 March 2023). The ExPASy server [29] (https://web.expasy.org/protparam/, accessed on 7 March 2023) was subsequently used to predict the physicochemical properties (e.g., number of amino acids, protein molecular weight (MW), theoretical isoelectric point (pI), instability index, aliphatic index, and grand average of hydropathicity (GRAVY)) of the *SmPHT1* proteins. The subcellular localization of the proteins was predicted using the Plant-PLoc server [30] (www.csbio.sjtu.edu.cn/bioinf/plant/, accessed on 8 March 2023). Finally, the amino acid sequences of the PHT1 proteins in *A. thaliana* and *O. sativa* were downloaded from the Arabidopsis Information Resource (TAIR) database (http://www.arabidopsis.org, accessed on 3 March 2023) and the rice genome database (http://rice.plantbiology.msu.edu/, accessed on 6 March 2023), respectively. A phylogenetic tree of PHT1 proteins was constructed in MEGA-X using the maximum likelihood (ML) method with 1000 bootstrap replicates [31].

### 2.3. Gene Structure, Conserved Motif, and Cis-Acting Element Analysis

The exon/intron distribution of each *SmPHT1* gene was analyzed using the Gene Structure Display Server [32] (GSDS, http://gsds.cbi.pku.edu.cn/, accessed on 14 March 2023). The conserved motifs were analyzed using Multiple EM for Motif Elicitation [33] (MEME, https://meme-suite.org/meme/, accessed on 15 March 2023), with a maximum of 15 motifs per sequence. After cis-acting regulatory element analysis, the 2000 bp upstream sequence of the transcription initiation site was analyzed using the Plant Care website [34] (https://bioinformatics.psb.ugent.be/webtools/plantcare/html/, accessed on 18 March 2023).

### 2.4. RNA Extraction and SmPHT1 Gene Expression

Total RNA was extracted from the samples using the FastPure Plant Total RNA Isolation Kit (Vazyme, Nanjing, China) and subsequently reverse-transcribed into complementary DNA (cDNA) (TaKaRa, Dalian, China). RT-qPCR analysis was performed using a CFX96 Real-Time System (Bio-Rad, Hercules, CA, USA) with TB Green Premix Ex Taq II (TaKaRa, Dalian, China). Primers for RT-qPCR were designed using Primer3 based on the CDS of the genes, and the NCBI database was used to determine primer specificity. The *β-actin* gene was chosen as the reference gene. The sequences of the primers used in this study are listed in Appendix A. Relative expression was calculated using the 2^−ΔΔCt^ method [35].

### 2.5. Analysis of Mycorrhizal Colonization

The root of *S. miltiorrhiza* were cut into segments approximately 1 cm in length. Then, the root segments were placed into test tubes, and Trypan blue staining was performed according to the methods of Phillips and Hayman [36]. First, the root segments were immersed in a 10% KOH solution and heated in a water bath at 90 °C for 40 min. Next, the segments were immersed in a 2% HCl solution for 5 min, 0.05% phenol cotton blue staining solution was added, and the test tubes were heated in a water bath at 90 °C for 30 min. Finally, after rinsing with distilled water, an appropriate amount of decolorizing solution (water–glycerol = 1:1) was added for 24 h of decolorization treatment, after which the results were observed. The mycorrhizal colonization rate was calculated via light microscopy using the gridline intersection method [37].

### 2.6. Data Statistics and Analysis

The statistical data were analyzed using SPSS software and Student’s *t* test. All experiments were repeated in triplicate. Differences were considered significant at *p* < 0.01 and *p* < 0.05.

## 3. Results

### 3.1. Identification and Phylogenetic Relationships of SmPHT1

We identified nine *SmPHT1* genes from the *S. miltiorrhiza* genome (Table 1). The amino acid sequence length of *SmPHT1* ranged from 516 aa (SmPHT1;4) to 542 aa (SmPHT1;7), with an average length of 527 aa. The molecular weight ranged from 55,704.84 to 59,451.05 Da, and the theoretical pI ranged from 8.32 (SmPHT1;1) to 9.15 (SmPHT1;8). Except for SmPHT1;4, the instability indices of the remaining proteins were less than 40, indicating that all the proteins were stable (S). The hydrophobicity of the nine PHT1 proteins was positive, indicating that they are hydrophobic proteins. Subcellular localization prediction revealed that all the *SmPHT1* proteins were located on the plasma membrane.

To investigate the phylogenetic relationships and functional associations, phylogenetic trees were constructed from *S. miltiorrhiza* (*Sm*), *A. thaliana* (*At*), and *O. sativa* (*Os*). The PHT1 proteins were clustered into three subgroups: Group I, Group II, and Group III (Figure 1). There were six *SmPHT1* (SmPHT1;1, SmPHT1;2, SmPHT1;5, SmPHT1;7, SmPHT1;8, SmPHT1;9) proteins in Group I, two SmPHT1 (SmPHT1;3, SmPHT1;6) in Group II, and one *SmPHT1* (SmPHT1;4) in Group III.

### 3.2. Gene Structures and Conserved Motifs

To understand the structural diversity of *SmPHT1*, exons and introns were further analyzed (Figure 2A). According to the results of the gene structure analysis, the number of exons varied from 1 to 2. The *SmPHT1;1*, *SmPHT1;3*, *SmPHT1;5*, *SmPHT1;6*, *SmPHT1;7*, and *SmPHT1;9* genes contained no introns, while the *SmPHT1;2*, *SmPHT1;4*, and *SmPHT1;8* genes contained 1 intron.

To further investigate the structural features of *SmPHT1*, a total of 15 putative motifs were identified (Figure 2B). The number of conserved domains in each protein ranged from 10 to 15. Motif 1 was present in all the *SmPHT1* proteins detected in *S. miltiorrhiza* (Figure 2C). Motifs 3 and 8 were located in the N- and C-terminal domains of *SmPHT1*, respectively. Motifs 1–7 were present in all the SmPHT1 proteins. Motif 15 was detected only in SmPHT1;2, SmPHT1;3, and SmPHT1;4. In addition, SmPHT1;3 lacked motif 12; SmPHT1;4 lacked motif 8, motif 10, motif 13, and motif 14; and SmPHT1;6 lacked motif 9 and motif 15. These differences in motif components might lead to different gene functions.

### 3.3. Cis-Acting Element Analysis

To understand the transcriptional regulation and potential functional roles of *SmPHT1*, the region 2000 bp upstream of the initiation codon was analyzed. Among the PHT1 promoters, seven hormone-related, six stress-related, six development-related, and twenty-two light-related elements were identified (Appendix A). Among the hormone-related elements, ABREs (ABA responsiveness), CGTCA motifs (MeJA responsiveness), TGACG motifs (MeJA responsiveness), and TCA elements (SA responsiveness) were found in all *SmPHT1* promoters. The gibberellin-responsive element was found only in *SmPHT1;3* (TATC-box) and *SmPHT1;9* (P-box). Sp1 (drought inducibility), Box4 (light responsiveness), G-Box (light responsiveness), and G-box (light responsiveness) were also observed in the promoter regions of all the *SmPHT1s*. However, *SmPHT1;9* lacked any elements associated with development-related genes. These results implied that the *SmPHT1* genes could be involved in hormone signal responsiveness and stress adaptation.

### 3.4. Mycorrhizal Colonization

To investigate the symbiotic relationship between AMF and the roots of *S. miltiorrhiza* under different P concentrations, mycorrhizal colonization was analyzed. In the absence of AMF, the mycorrhizal colonization structure of *S. miltiorrhiza* was not established (Figure 3A). After inoculation with AMF, the colonization rate was greater at 0.1 mM P than 0 mM and 10 mM P (Figure 3B). These results indicated that a P concentration of 0.1 mM significantly increased the colonization rate of the AMF.

### 3.5. Effect of AMF Colonization on the Growth and P uptake of S. miltiorrhiza

To study the effects of AMF colonization on the growth and nutrient absorption of *S. miltiorrhiza*, the biomass of *S. miltiorrhiza* plants and the total P content in the roots were measured. Compared with those in the non-AMF inoculation treatment, the *S. miltiorrhiza* plants that were inoculated with AMF had significantly greater root biomass at 0 mM and 0.1 mM P (Figure 4A,B). Moreover, the total P content in the roots of AMF-inoculated plants was significantly greater than that in the roots of non-AMF-inoculated plants under the 0.1 mM P concentration (Figure 4C). However, no significant differences in either the biomass or P content were observed between the AMF-inoculated and non-AMF plants under high P supply conditions (Figure 4B,C). The results indicated that AMF inoculation significantly promoted *S. miltiorrhiza* plant growth and P absorption.

### 3.6. Tissue-Specific Expression Patterns of SmPHT1

To investigate the potential biological functions of *SmPHT1* genes, the expression patterns of the *PHT1* genes in the roots, stems, leaves, and flowers were analyzed. As shown in Figure 5, *SmPHT1;7* and *SmPHT1;8* were highly expressed in the roots; *SmPHT1;1*, *SmPHT1;3*, *SmPHT1;4*, and *SmPHT1;6* were highly expressed in the stems; *SmPHT1;2* was highly expressed in the leaves; and *SmPHT1;5* and *SmPHT1;9* were highly expressed in the flowers. Based on the results of the organizational expression pattern analysis, the *PHT1* genes in *S. miltiorrhiza* were shown to be expressed mainly in roots and stems and less expressed in leaves and flowers.

### 3.7. Expression Patterns of SmPHT1 Genes in Response to P and AMF

To further investigate the functional role of *SmPHT1* genes in *S. miltiorrhiza*, their expression levels were examined in *S. miltiorrhiza* roots inoculated with AMF under different P concentrations (Figure 6). First, with increasing P concentration, there was a significant decrease in the expression of most *SmPHT1* genes between the colonized and the noncolonized plants. At a P concentration of 10 mM with or without AMF colonization, *SmPHT1;5* and *SmPHT1;8* expression was almost absent. AMF inoculation under P-free conditions could induce *SmPHT1;5* and *SmPHT1;9* expression, but they were suppressed when 0.1 mM P was added. Moreover, there were no significant differences in the expression levels of *SmPHT1;2*, *SmPHT1;3*, *SmPHT1;4*, *SmPHT1;7*, or *SmPHT1;8* between the colonized and the noncolonized plants. Notably, the expression level of *SmPHT1;6* was significantly greater in the AMF-colonized plants than in the non-AMF-colonized plants. The present study provides a valuable description of *SmPHT1* gene expression regulation during P stress and AMF colonization.

## 4. Discussion

The PHT1 family plays a crucial role in the uptake and transport of P from the soil to ensure the optimal growth and development of plants. Investigating the specificity and biochemical characteristics of PHT1 will contribute to a better understanding of P homeostasis and utilization efficiency in plants [38]. In the present study, we conducted a comprehensive analysis of *SmPHT1* genes in *S. miltiorrhiza*. A total of nine *SmPHT1* genes were identified in the *S. miltiorrhiza* genome. Nine *PHT1* genes were found in *A. thaliana* [39], thirteen *PHT1* genes were found in *O. sativa* [18], six *PHT1* genes were found in *Astragalus sinicus* [40], twenty-one *PHT1* genes were found in *Triticum aestivum* [41], and forty-nine *PHT1* genes were found in *Brassica napus* [42]. The variation in the number of *PHT1* genes among different species may be attributed to the occurrence of gene duplications and gene loss during evolution [43,44]. Gene duplication followed by functional differentiation has played a crucial role in driving evolutionary novelty, enabling plants to increase their adaptability to new environments [45]. The results of this study provide information for the functionality of the *PHT1* gene in *S. miltiorrhiza* under P stress.

Phylogenetic analysis serves as an important tool in comparative research on any biological process. Phylogenetic analysis of *SmPHT1* proteins in *S. miltiorrhiza*, together with those of *A. thaliana* and *O. sativa*, revealed that PHT1s can be classified into three main groups. Similar results have also been found in *Gossypium hirsutum* [46]. These results also demonstrated that PHT1 was unevenly distributed among the groups and the PHT1 gene family members in the *S. miltiorrhiza*, *A. thaliana*, and *O. sativa* subfamilies. Consistent with the current information on plant evolution, the phylogenetic tree indicated that *SmPHT1* was more closely related to its counterparts in *A. thaliana* (eudicot) than to its counterparts in *O. sativa* (monocot) (Figure 1). Phylogenetic analysis of PHT1s revealed similar evolutionary divergences that reflected differences in biochemical and functional properties [7]. The distribution patterns of PHT1 proteins suggested that the *SmPHT1* protein could perform similar biological processes as its counterparts in *A. thaliana* or *O. sativa*.

PHT1 proteins have a conserved structure containing 12 transmembrane (TM) domains with a large hydrophilic loop between TM6 and TM7, and the C- and N-termini are expected to be oriented inside the cell, with the protein inserted in the plasma membrane [47,48]. PHT1 has the conserved feature sequence GGDYPLSATIxSE [49]. PHT1 in plants is responsible for the expression of H+/Pi symporters, which exhibit similarities in sequence, structure, and size across various plant species [50]. Our present findings align with the existing knowledge on this subject. Motif analysis revealed that the domain was located in motif 1 of *SmPHT1* and was present in all the PHT1s in *S. miltiorrhiza*. Furthermore, genes within the same branch exhibited a similar distribution of basic sequences (Figure 2C). Similar results were found in tea plants (*Camellia sinensis*) [51] and tomatoes (*Solanum lycopersicum*) [7]. These conserved motifs possibly have functional and/or structural roles in active proteins [52]. Analysis of the gene structures indicated that the numbers and distributions of exons and introns within each branch were relatively similar. Of the 20 coding sequences of *SmPHT1* genes, 6 contained no introns, and the rest were destroyed by introns [10]. The variations in motifs and structures between different branches demonstrated the diversity in the functionality of the *SmPHT1* gene family in *S. miltiorrhiza*. These findings indicate that the *SmPHT1* gene family is relatively conserved throughout the process of evolution, guiding subsequent functional research.

Most members of the PHT1 family exhibit strong expression in the roots, regardless of whether they are found in dicotyledonous or monocotyledonous plants [12,40,53,54]. However, the expression of the *PHT1* gene has also been detected in other organs, such as leaves, flowers, seeds, fruits, and aleurones [11,55,56]. To investigate the functions of these *SmPHT1* genes in more detail, we examined the transcriptional patterns of these genes in the roots, stems, leaves, and flowers of *S. miltiorrhiza*, which could provide important clues about their functions. In the present study, the expression of *SmPHT1* varied to different degrees across the roots, stems, leaves, and flowers of *S. miltiorrhiza* (Figure 5), where it may play a role in P uptake and translocation. *SmPHT1;1*, *SmPHT1;3*, *SmPHT1;4*, *SmPHT1;7*, and *SmPHT1;8* were highly expressed in the stem and root, implying that they may play roles in P absorption and transport [43,57,58]. Similarly, in *Gossypium hirsutum*, the expression of *GhPHT1;3-At*, *GhPHT1;4-At*, *GhPHT1;5-At*, *GhPHT1;3-Dt*, *GhPHT1;4-Dt*, and *GhPHT1;5-Dt* is increased in the roots and stems [46]. *SmPHT1;2*, *SmPHT1;5*, and *SmPHT1;9* were highly expressed in leaves and flowers. The transcript levels of *LbPT1*, *LbPT2*, and *LbPT7* in *Lycium barbarum* were highest in senescent leaves [59], and *SgPT3* in *Stylosanthes guianensis* was highly expressed in flowers [60], indicating the potential role of these genes in reproductive growth [46]. These variable expression patterns indicate that the *PHT1s* involved in development and other physiological processes are functionally diverse.

AMF are among the most widely distributed symbiotic organisms in plant roots. The existence of AMF symbiosis in many early diverging land plants, including liverworts, hornworts, lycophytes, and ferns, reveals that AMF symbiosis predates the development of true root systems. AMF symbiosis provides a pathway for the acquisition of P and other mineral nutrients [60]. Studies have shown that the development of AMF is influenced by the availability of P in the environment and by the P transport proteins within plants [10,52]. Moreover, the extent of symbiotic associations between different types of AMF and their hosts substantially influences P absorption by mycorrhizae. For *Sorghum bicolor*, mycorrhizal colonization by *Rhizophagus irregularis* and *Funneliformis mosseae* was only visible in the treatments in which plants were inoculated with AMF, ranging from 52% to 93% for arbuscular colonization [61]. Similar results were found in *Petunia hybrida*; with increasing P concentration, AMF colonization was inhibited and was already very low at a concentration of 10 mM P [62]. In the present study, AMF colonization was inhibited by 10 mM P (Figure 3B). Our findings indicate that AMF play an important role in plant P uptake.

The inoculation of AMF can promote the adaptability of host plants, especially by providing additional P and improving plant growth [63]. *S. miltiorrhiza*, a mycotrophic plant, readily forms symbiotic associations with AMF [26]. Comparisons of the growth status of *S. miltiorrhiza* between the AMF-inoculated and non-AMF-inoculated plants under different P supply conditions showed that the symbiotic relationship of AMF significantly promoted the growth of *S. miltiorrhiza* (Figure 4A). In addition, under low-P conditions (0.1 mM), inoculation with AMF significantly increased the fresh weight and P content of the roots (Figure 4B,C), which is consistent with the findings of previous studies of *S. lycopersicum* [43] and *G. hirsutum* [63].

The symbiotic relationship between AMF and plants can induce the expression of PHT1 under P stress, as observed in Hordeum vulgare [53], Glycine max [64], and *S. bicolor* and *Linum usitatissimum* [65]. In the present study, the transcript levels of *SmPHT1* in *S. miltiorrhiza* roots were examined after AMF inoculation under different P concentrations (0 mM, 0.1 mM, and 10 mM). These findings indicated that the *SmPHT1* gene was differentially expressed in response to symbiotic AMF and P stress. The expression of *SmPHT1* was significantly repressed under high-P conditions. PHT1 reportedly responds to P stress, as observed in *S. lycopersicum* [43] and *Zea mays* [66]. In addition, PHT1s, such as EgPT8 in *R. irregularis* [67], AsPT1 in *A. sinicus* [40], and ZEAma: Pht1;6 in *Z. mays* [68], are specifically induced during mycorrhization and have been found to be crucial for mycorrhizal P uptake. Notably, in the present study, the expression of SmPHT1;6 was markedly greater in AMF-inoculated roots than in non-inoculated roots. Furthermore, SmPHT1;6 exhibited orthologous relationships with OsPHT1;11 (Figure 1). OsPHT1;11 plays a role in the mycorrhizal symbiosis process, and its expression is induced by mycorrhizal infection; it is responsible for the uptake of inorganic P from mycorrhizal fungi [69]. Similarly, ZmPHT1;6 clustered with OsPHT1;11 and was found to be a specific AMF-inducible PHT1 [66]. Therefore, SmPHT1;6 may play an important role in mycorrhizal symbiosis and P absorption.

## 5. Conclusions

In the present study, a total of nine *SmPHT1* genes were identified. The phylogenetic relationships, gene structures, conserved motifs, and cis-acting elements of the genes were analyzed, and the results provided insights into their biological functions. Furthermore, tissue expression analysis revealed that most of the *SmPHT1* genes were expressed in the roots and stems. *SmPHT1* genes were differentially expressed upon inoculation with AMF under P stress. Notably, *SmPHT1;6* expression was strongly induced by AMF symbiosis. These results provide a useful foundation for future work and could lead to a better understanding of the evolutionary regulatory mechanisms of PHT1 genes in response to P stress and AMF symbiosis in *S. miltiorrhiza*.

## Figures and Tables

**Figure 1 genes-15-00589-f001:**
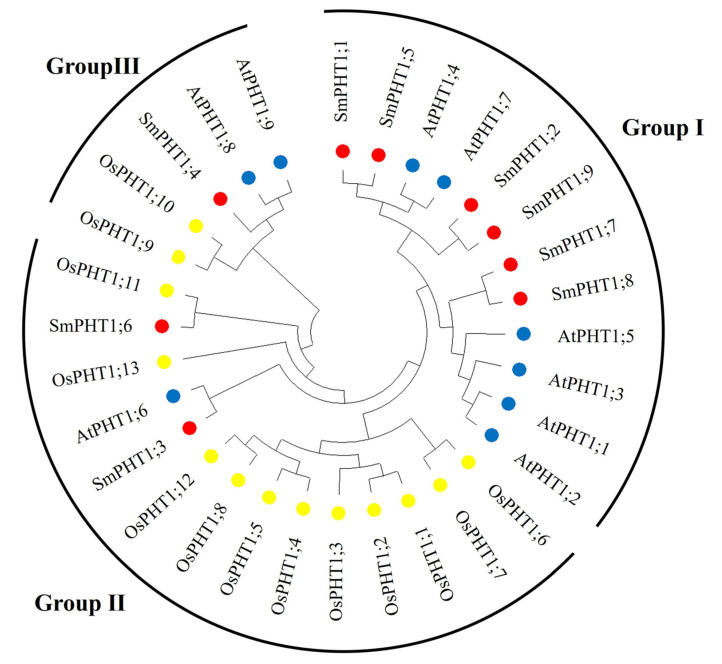
Phylogenetic tree of the PHT1 proteins from *S. miltiorrhiza*, *A. thaliana*, and *O. sativa*. Proteins from *S. miltiorrhiza*, *A. thaliana*, and *O. sativa* are denoted by red circles, blue circles, and yellow circles, respectively. Bootstrap values was 27–100. The sequences used in this study are listed in Appendix A.

**Figure 2 genes-15-00589-f002:**
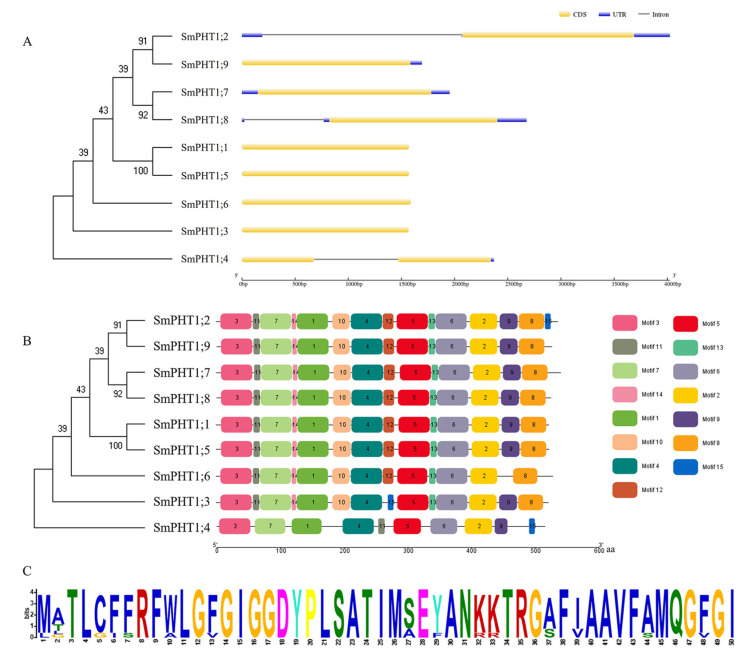
Conserved motifs and gene structures of the *SmPHT1* in *S. miltiorrhiza*. (**A**) Gene structures. The yellow boxes indicate exons, the blue boxes indicate untranslated 5′ and 3′ regions, and the black lines indicate introns. (**B**) Motif composition. The motifs are displayed in different colored boxes. The sequence information for each motif is provided in Appendix A. (**C**) Visualization of motif 1.

**Figure 3 genes-15-00589-f003:**
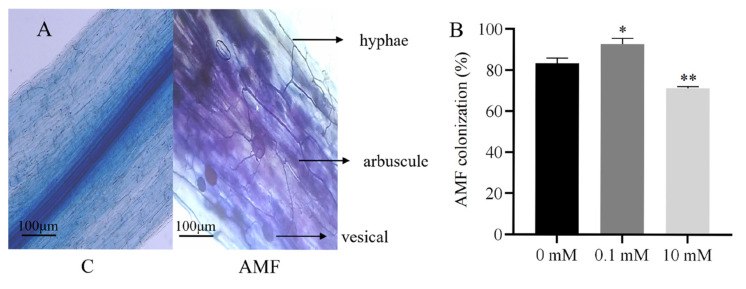
Impact of external P concentration on AMF symbiosis in *S. miltiorrhiza* roots: (**A**) typical structure of AMF in the root system of *S. miltiorrhiza*; (**B**) percentage of total mycorrhizal colonization in *S. miltiorrhiza* plants. * *p* < 0.05, ** *p* < 0.01.

**Figure 4 genes-15-00589-f004:**
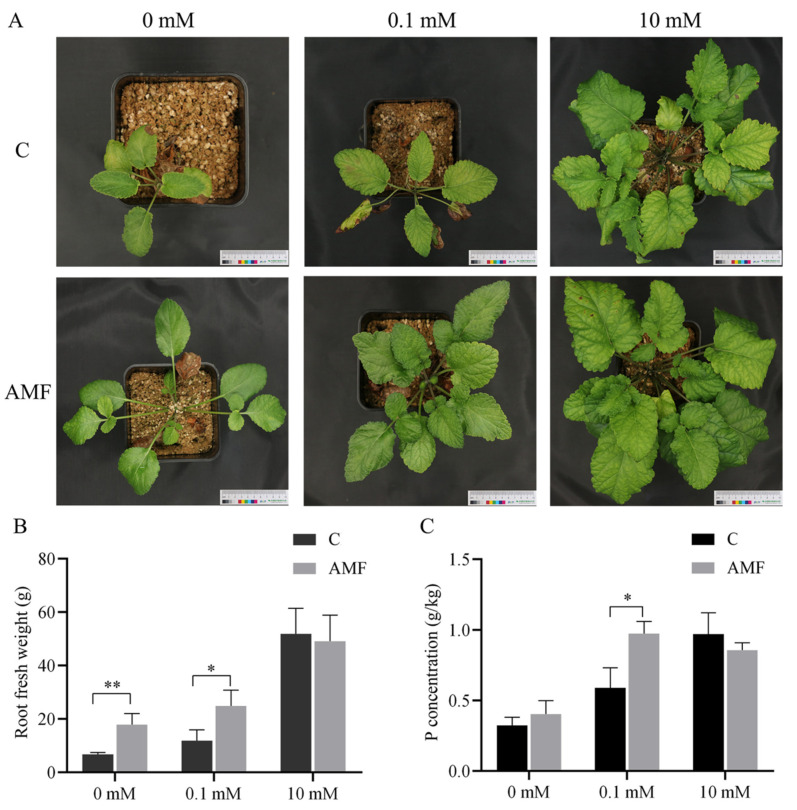
The impact of AMF colonization on the growth of *S. miltiorrhiza* and P absorption was investigated under the influence of different P concentrations: (**A**) growth status of *S. miltiorrhiza* under different P concentrations with AMF colonization; (**B**) fresh root biomass; (**C**) root total P content. The asterisks indicate significant differences. C stands for non-mycorrhizal. * *p* < 0.05, ** *p* < 0.01.

**Figure 5 genes-15-00589-f005:**
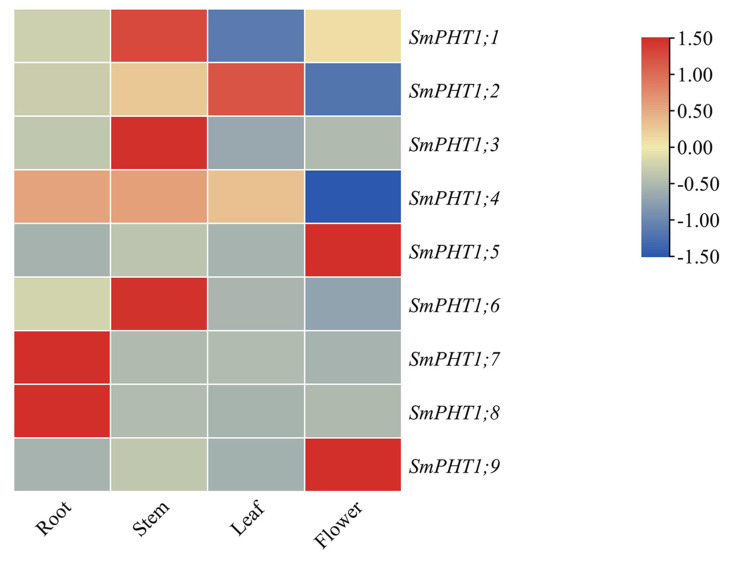
Expression analysis of *SmPHT1* genes in *S. miltiorrhiza*. Heatmap of *SmPHT1* expression in the roots, stems, leaves, and flowers. The color scale limits were artificially set to −1.50 to 1.50 according to the normalized values. The gradient from blue to red represents low to high expression.

**Figure 6 genes-15-00589-f006:**
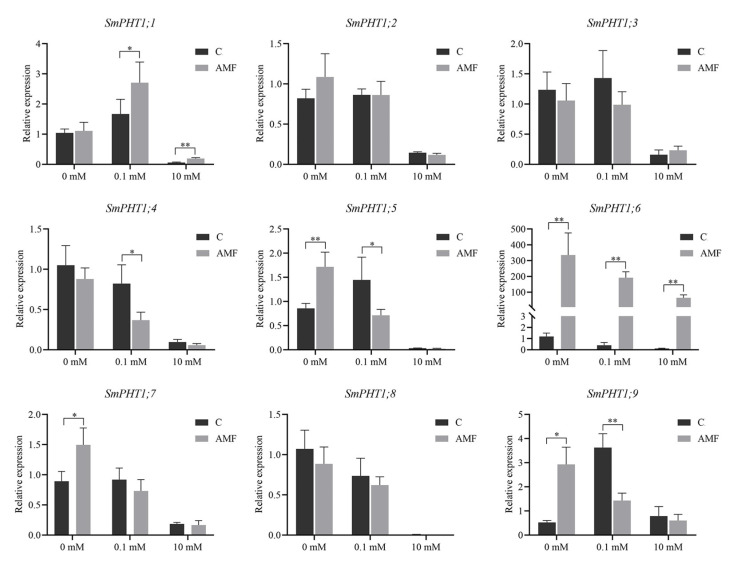
Expression patterns of *SmPHT1* genes under P and AMF in *S. miltiorrhiza*. Transcript profiling of the root tissues of AMF and non-AMF plants. *SmActin* was used as an endogenous control. The asterisks indicate significant differences. * *p* < 0.05, ** *p* < 0.01.

**Table 1 genes-15-00589-t001:** Characteristics of *SmPHT1* identified from *S. miltiorrhiz*.

Name	Amino Acids	Molecular Weight (Da)	Theoretical pI	Instability Index	Grand Average of Hydropathicity	Subcellular Localization
SmPHT1;1	523	57,218.7	8.32	30.35 S	0.376	Plasma membrane
SmPHT1;2	537	58,947.83	8.86	32.44 S	0.312	Plasma membrane
SmPHT1;3	522	56,466.87	9.01	35.33 S	0.417	Plasma membrane
SmPHT1;4	516	55,704.84	8.83	43.73 US	0.443	Plasma membrane
SmPHT1;5	523	57,184.76	8.64	30.96 S	0.37	Plasma membrane
SmPHT1;6	529	58,482.94	8.67	32.11 S	0.249	Plasma membrane
SmPHT1;7	542	59,451.05	8.98	31.81 S	0.281	Plasma membrane
SmPHT1;8	525	57,433.07	9.15	30.96 S	0.336	Plasma membrane
SmPHT1;9	527	57,588.11	9.07	38.79 S	0.32	Plasma membrane

## Data Availability

All data analyzed during this study are included in this article and its additional files.

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
