# Peer review of "Genome-Wide Identification and Characterization of the PHT1 Gene Family and Its Response to Mycorrhizal Symbiosis in Salvia miltiorrhiza under Phosphate Stress"

_genes, 2024, doi:10.3390/genes15050589_

Round 1

Reviewer 1 Report

Comments and Suggestions for Authors

Article title:Genome-wide identification and characterization of the PHT1 gene family and its response to Mychorrhizal symbiosis in Salvia
miltiorrhiza under phosphate stress.

Authors: Chen et al.,

General comments:

The manuscript describes a correlational/in silica/gene expression study on the nature of gene influencing phosphate transport in the medicinal model plant Salvia miltiorrhiza.

The design is sounds and the data are presented with a balanced approach. The study does provide some insights in some of the
transporter genes however lacks any in depts and is very observational in nature.

Specific comments

Line 36. Sell out P when starting a sentence.
Line 39. Remove the from “the P supply”.
Line 95, provide a ratio for the components of the inoculum.
Line 96: clarify the inactivated inoculum.
Line 141. Change infection to colonization.
Line 155. The symbol for P value is capital and non-italicized. Change
throughout the manuscript.
Table 1. Indicate the units for the MW (Da). Instability index, use S for
stable and unS for unstable thus replacing the full word.
Figure 2 B clarify what are those motifs. Expand in the relevant results’
sections the nature and meaning of those motifs.
Figures 3, 4 and 6. Change CK to C for the control group.
Lines 226 and 229, indicate in brackets the P values.

Reviewer 2 Report

Comments and Suggestions for Authors

The manuscript “Genome-Wide Identification and Characterization of the PHT1 2 Gene Family and its Response to Mycorrhizal Symbiosis in Salvia miltiorrhiza under Phosphate Stress” by Bai et al. describes a genome wide characterization of the PHT1 gene family of phosphate transporters in the medicinal plant Salvia miltiorrhiza. This is in part a bioinformatics study, but als includes qRT-PCR of all members of this family in S. miltiorrhiza. The PHT1 family is an interesting family in regard to phosphate uptake, mycorrhizal colonization and P translocation, and thus is of general interested for adaptation to P deficiency. The paper further explores the effect of different P concentrations and arbuscular mycorrhizal fungi (AMF) on plant growth and PHT1 gene expression. While I agree that this is an important topic for crop plants, It would be helpful to further explain why this is an important topic for this medicinal plant; e.g. is P deficiency an issue for growing S. miltiorrhiza?

The English language throughout this paper is excellent. But I often found that more specifics would help to better understand and interpret the presented results; I listed specific points below.

INTRODUCTION

“P primarily exists in the soil in complex, insoluble organic forms”

What about inorgaic P?

S. miltiorrhiza … is considered a model organism for medicinal plant research.”  Add  citation to support this claim

“The gene expression patterns in response to different P levels (0 mM, 86 0.1 mM, and 10 mM) during AMF colonization were investigated.” As a reader, I would already like to know how; I suggest to add “by RT-qPCR”.

MATERIALS AND METHODS

“S. miltiorrhiza plants with consistent growth” What does that mean? How old were these plants; how big (volume) were the pots? 

“with modified Hoagland solution” How was it modified? Was P solution replaced with any solution?

“and the NCBI database was used to determine primer specificity” Which NCBI database?

“The β-actin gene was chosen as the reference gene.” Based on what? Do you know if this is suitable for the conditions you are testing? If so, state this.

“using the 2–ΔΔC_t_ _method” state if amplification efficiency been determined and taken into account

“The data were analyzed by one-way analysis of variance (ANOVA)…” Which data were analytzed via ANOVA; which data via ttest?

RESULTS

“Figure 1. Phylogenetic tree”

This is a nice tree, but bootstrap values or any other form of confidence indicators should be added.

“After inoculation with AMF, the colonization rate was greater at 0.1 mM…” remind the reader how many biological replications (independent plants) were analyzed for each P concentration.

“Figure 4.” Again, mention the number of biological (independent) replications.

Figure 5

“The color scale limits were artificially set to -1.50 to 1.50 according to the normalized values.” Add some explanation what these numbers indicate. What normalized values? 

“Figure 6.” How many biological replications?

DISCUSSION

“This approach is highly important for studying the functionality..” What approach? The sentence before describes gene duplications, which is not an experimental approach.

“A. thaliana (dicot)” should be “eudicot”, “dicot” is no longer correct.

“The differences in the PHT1 gene, while exhibiting significant overlap in expression, effectively demonstrated the evolutionary conservation …” how do differences demonstrate conservation?

“AMF symbiosis occurs prior to true root development, such as in mosses, liverworts, ferns, and horsetails” 

True ferns and horsetails have true roots.

Reviewer 3 Report

Comments and Suggestions for Authors

Dear editors and authors, I have read and reviewed the manuscript entitled Genome-Wide Identification and Characterization of the PHT1 2 Gene Family and its Response to Mycorrhizal Symbiosis in Salvia miltiorrhiza under Phosphate Stress, authored by Yanhong Bai, Yanan Lin, Hongyan Liu, Fengxia Han, Hui Chang, Menglin Li, Xue Chen and Qian Liu.

The following are some suggestions for further revisions:

1.                  Describe the origin of the S. miltiorrhiza plants.

2.                  Describe the inoculum preparation (for example, chemical and physical characteristics, mix procedure).

3.                  How did you get the inactivated inoculum?

4.                  Define “CK” in figure 4.

5.                  There is an error in reference #44.

6.                  Explain how cis acting elements could regulate stress response by P in S. miltiorrhiza.

7.                  Based on your results, could you suggest possible molecular mechanisms or phosphate uptake signaling that regulate the PHT1 expression during mycorrhizal symbiosis under P stress in S. miltiorrhiza
